# Profiling of Breast Cancer Stem Cell Types/States Shows the Role of CD44^hi^/CD24^lo^-ALDH1^hi^ as an Independent Prognostic Factor After Neoadjuvant Chemotherapy

**DOI:** 10.3390/ijms26178219

**Published:** 2025-08-24

**Authors:** Hazem Ghebeh, Jumanah Y. Mirza, Taher Al-Tweigeri, Monther Al-Alwan, Asma Tulbah

**Affiliations:** 1Cell Therapy and Immunobiology Department, King Faisal Specialist Hospital & Research Centre, P.O. Box 3354, Riyadh 11211, Saudi Arabia; jmirza@kfshrc.edu.sa (J.Y.M.); malwan@kfshrc.edu.sa (M.A.-A.); 2College of Medicine, Al-Faisal University, Riyadh 11533, Saudi Arabia; 3Oncology Centre, King Faisal Specialist Hospital & Research Centre, Riyadh 11211, Saudi Arabia; ttwegieri@kfshrc.edu.sa; 4Department of Pathology, King Faisal Specialist Hospital & Research Centre, Riyadh 11211, Saudi Arabia; tulbah@kfshrc.edu.sa

**Keywords:** cancer stem cells, PD-L1, CD44^hi^/CD24^lo^, CD24, Ep-CAM, ALDH1, CD10, BMI1

## Abstract

Multiple markers exist for breast cancer stem cells (CSCs), which are believed to represent the phenotypes of various CSC types and/or states. The relationship between each CSC subpopulation/state and the primary hallmarks of cancer has not been sufficiently clarified. In this study, six CSC markers (CD44^hi^/CD24^lo^, CD24, Ep-CAM, ALDH1, CD10, and BMI1) were assessed in a surgical cohort of 73 breast cancer patients. The expression of a single or multiple CSC markers was correlated with clinicopathological parameters, including markers of immune evasion, proliferation, epithelial–mesenchymal transition (EMT), and survival. All CSC phenotypes, except for CD10, correlated with markers indicative of higher proliferation. The CD44^hi^/CD24^lo^ phenotype correlated with markers of EMT and PD-L1 expression, unlike ALDH1^hi^. Both Ep-CAM^hi^ and CD24^hi^ breast cancer were associated with indicators of immune evasion, including PD-L1 expression, and the infiltration of FOXP3+ and PD-1+ tumor-infiltrating lymphocytes (TIL). While the CD44^hi^/CD24^lo^, Ep-CAM^hi^, and ALDH1^hi^ phenotypes correlated with shorter overall survival (OS), CD24^hi^ correlated with reduced disease-free survival (DFS). Interestingly, among all tested CSC markers, the CD44^hi^/CD24^lo^-ALDH1^hi^ combination phenotype correlated with the worst DFS (HR 2.8, *p* = 0.014 in univariate/multivariate analysis) and OS (*p* < 0.001, HR 6.4 in univariate and 5.4 in multivariate analysis). A side-by-side comparison of multiple CSC markers demonstrated the differential linkage of CSC phenotype/state with distinct features of breast cancer. This comparison demonstrates the advantage of the CD44^hi^/CD24^lo^-ALDH1^hi^ combination marker for prognostication, especially after neoadjuvant chemotherapy. In the future, distinct markers of CSCs can hopefully be leveraged to trace/monitor different disease characteristics or treatment outcomes.

## 1. Introduction

A subpopulation of cancer cells, known as cancer stem cells (CSCs), is responsible for maintaining and regenerating tumors. In principle, CSCs are therapy-resistant and have higher metastatic capabilities [1]. The CD44^hi^/CD24^lo^ phenotype and the ALDH1^hi^ trait are the most commonly used markers to identify CSCs in breast cancer [1]. Additionally, other markers for CSCs in breast cancer have been reported, including Ep-CAM^hi^ [2], the CD44^hi^/CD24^lo^-Ep-CAM^hi^ combination [3,4,5], CD10 [6], CD24^hi^ [7], and BMI1 [8]. While one could assume that CSCs are a single population with multiple markers, several reports have demonstrated the existence of different CSC subpopulations [9]. The demonstration of a dynamic relationship between these subpopulations [10] suggests the existence of diverse CSCs in different dynamic states.

The role of the immune response in the surveillance and elimination of cancer cells is well-established [11]. However, strong evidence indicates that CSCs can interact with immune cells differently from the bulk of the tumor cells [12]. Despite this, the specific CSC subpopulation or state correlating with immunosuppressive cells or environments remains understudied. We have previously shown that the immune checkpoint PD-L1 is overexpressed in CSCs and is correlated with CD44^hi^/CD24^lo^ in breast cancer cell lines [13,14]. In contrast, there was no correlation between PD-L1 expression and ALDH^hi^ cancer cells [14]. The CSC type or state linked to PD-L1 expression has not been sufficiently explored.

In this study, we utilized immunohistochemistry to characterize CSCs in breast cancer patients using multiple reported CSC markers. We investigated the correlation of each CSC type/state with markers of immune evasion (including PD-L1 expression and the infiltration of FOXP3+ and PD-1+ TIL), cell proliferation, and Epithelial-to-Mesenchymal Transition (EMT). Additionally, we evaluated the prognostic significance of these markers individually and in conjunction with other clinicopathological parameters. While CD44^hi^/CD24^lo^, Ep-CAM^hi^, and their combination (i.e., CD44^hi^/CD24^lo^-Ep-CAM^hi^) showed significant correlation with PD-L1 expression, a marker of immune evasion, ALDH1, CD10, or BMI1 did not. The combination of CD44^hi^/CD24^lo^-ALDH1^hi^ emerged as the most prognostic among all tested CSC markers. These markers could serve as valuable tools for monitoring disease progression during treatment, and the combined targeting of ALDH1^hi^ and CD44^hi^/CD24^lo^ phenotypes may represent the most effective strategy to eliminate CSCs and address breast cancer at its roots.

## 2. Results

### 2.1. Patient Characteristics

All patients were females with breast cancer and a median age of 43 years. A substantial proportion (47%) had large tumors (≥4 cm), and nearly half (49%) had histologic grade 3 tumors. The majority (64%) showed lymph node involvement. Most patients (63%) had luminal (A or B) subtype tumors, while 16% had the HER2-enriched subtype, and 21% had triple-negative breast cancer (TNBC) (Appendix A).

Among the cohort, 43 patients received neoadjuvant chemotherapy. Of these, 37 were treated with a combination of doxorubicin (Adriamycin^®^, Pfizer, New York, NY, USA) and cyclophosphamide (AC), while 6 received other regimens. Among the 37 patients who received AC, 21 also received fluorouracil as part of the FAC regimen, +/− docetaxel, 9 received only docetaxel following AC, while the remaining 7 received AC alone.

### 2.2. CD44^hi^/CD24^lo^ CSCs Phenotype Correlates with EMT, Proliferation, and PD-L1 Expression

Combined high expression of CD44 with low or absent expression of CD24 (CD44^hi^/CD24^lo^) is one of the most commonly used markers to identify CSCs in breast cancer [3]. The CD44^hi^/CD24^lo^ phenotype was positive in 33% of the tested cohort of breast cancer patients (Figure 1A). The CD44^hi^/CD24^lo^ phenotype partially correlated with markers of immune evasion, showing significance with PD-L1 expression (*p* = 0.001), but not with FOXP3+ or PD-1+ TIL (Table 1A). Moreover, the CD44^hi^/CD24^lo^ phenotype significantly correlated with Ki-67-positive status (*p* = 0.024) (Table 1A), high histological grade (*p* = 0.013), and estrogen receptor (ER) negative status (*p* = 0.002) (Table 1B). Notably, CD44^hi^/CD24^lo^ significantly correlated with markers of EMT, including vimentin overexpression (*p* < 0.001), loss of E-cadherin (*p* = 0.015), and the combination of vimentin overexpression and loss of E-cadherin (*p* < 0.001) (Table 1B).

Around half (55%) of tumors had cancer cells that were double positive for CD44 and CD24. There was no significant correlation between this double positive and any of the tested bad prognostic factors.

### 2.3. Ep-CAM^hi^ Phenotype Correlates with EMT, Proliferation, and Immune Evasion

Ep-CAM was overexpressed in 43% of the examined breast cancer cases (Figure 1A). Ep-CAM^hi^ cases significantly correlated with markers of immune evasion, namely PD-L1 expression (*p* = 0.019), FOXP3+ TIL (*p* = 0.012), and higher TIL (*p* = 0.024) in general, along with a borderline correlation with PD-1+ TIL (*p* = 0.052, Table 1A). There was a significant association between Ep-CAM^hi^ cells and markers of proliferation, namely Ki-67 (*p* = 0.018), loss of the cell cycle inhibitor p27 expression (*p* = 0.031), and the combined loss of p21 and p27 expression (*p* = 0.029). Furthermore, Ep-CAM^hi^ significantly correlated with larger tumor size (*p* = 0.036), negative status for ER (*p* = 0.008) and progesterone receptors (PR, *p* = 0.005), and TNBC status (*p* = 0.008) (Table 1B). Additionally, Ep-CAM^hi^ expression significantly correlated with breast cancer cases exhibiting combined vimentin overexpression and loss of E-cadherin (*p* = 0.045) (Table 1B), a strong feature of EMT.

We have further examined cases that had combined overexpression of Ep-CAM and the CD44^hi^/CD24^lo^ phenotype (CD44^hi^/CD24^lo^-Ep-CAM^hi^), which was present in 16% of the studied cases. CD44^hi^/CD24^lo^-Ep-CAM^hi^ significantly correlated with PD-L1 expression (*p* = 0.008) and the proliferation marker Ki-67 (*p* = 0.024) (Table 1A). In addition, the CD44^hi^/CD24^lo^-Ep-CAM^hi^ phenotype significantly correlated with the negativity status of ER (*p* = 0.007) and PR (*p* = 0.009), and TNBC status (*p* = 0.002) (Table 1B). Importantly, there was a significant correlation between CD44^hi^/CD24^lo^-Ep-CAM^hi^ and markers of EMT, as demonstrated by vimentin overexpression (*p* = 0.005), loss of E-cadherin (*p* = 0.005), and cases with combined vimentin overexpression and loss of E-cadherin (*p* = 0.002) (Table 1B).

Altogether, CD44^hi^/CD24^lo^, Ep-CAM overexpression, and/or their combination (CD44^hi^/CD24^lo^-Ep-CAM^hi^) correlated with PD-L1 expression, higher proliferation, EMT, and ER-negative status.

### 2.4. CD24 Overexpression Correlates with Markers of Immune Evasion and Higher Proliferation

CD24 was expressed in the luminal epithelial cells of the normal-like breast ducts. CD24 was overexpressed in 34% of the tested breast cancer cases (Figure 1B). Interestingly, CD24 overexpression significantly correlated with markers of immune evasion including PD-L1 (*p* = 0.013), FOXP3+ CD3 + TIL (*p* = 0.001), and PD-1+ TIL (*p* = 0.032) as well as markers of proliferation Ki-67 (*p* = 0.048) and SKP2 (*p* = 0.002) expression, and the combined loss of p21/p27 (*p* = 0.014) (Appendix A). Moreover, CD24 overexpression significantly correlated with high histological grade (*p* = 0.007), ER-negative status (*p* = 0.005), and vimentin overexpression (*p* = 0.030) (Appendix A).

### 2.5. ALDH1^hi^ Phenotype Correlates with Higher Proliferation

ALDH1 expression is another very important marker that is used to identify CSCs in breast cancer. In our cohort of breast cancer patients, ALDH1 was mostly expressed by the mesenchymal cells and some normal mammary glands with hyperplasia. ALDH1 expression in tumor cells was observed in 30% of the breast cancer cases (Figure 1B). ALDH1 expression correlated with markers of higher proliferation, including loss of p27 (*p* = 0.041), and borderline correlation with Ki-67 (*p* = 0.074) (Table 2A), larger tumor size (*p* < 0.001), and higher histological grade (*p* = 0.011) (Table 2B). There was no significant correlation between ALDH1^hi^ and PD-L1 expression or other markers of immune evasion (Table 2A). Interestingly, there was a correlation between ALDH1^hi^ and higher TIL (*p* = 0.015).

Cases that had combined overexpression of the CD44^hi^/CD24^lo^ phenotype and ALDH1^hi^ (CD44^hi^/CD24^lo^-ALDH1^hi^) were observed in 11% of examined cases. Similar to what has been observed with ALDH1^hi^ alone, there was no significant correlation between CD44^hi^/CD24^lo^-ALDH1^hi^ and PD-L1 expression or other markers of immune evasion (Table 2A). However, it showed a significant correlation with larger tumor size (*p* = 0.022) and higher histological grade (*p* = 0.028) (Table 2B). CD44^hi^/CD24^lo^-ALDH1^hi^ significantly correlated with ER-negative status (*p* = 0.045) and vimentin overexpression (*p* = 0.049) (Table 2B).

On the other hand, having a combined Ep-CAM^hi^ and ALDH1^hi^ phenotype constituted 15% of the breast cancer cases. Like ALDH^hi^ tumors, Ep-CAM^hi^/ALDH1^hi^ breast cancers significantly correlated with markers of higher proliferation, including Ki-67 positivity status, combined loss of p21/p27 (*p* = 0.002) (Table 2A), and larger tumor size (*p* = 0.019) (Table 2B) in addition to ER negative status (*p* = 0.015). Similarly, Ep-CAM^hi^/ALDH1^hi^ cases did not correlate with PD-L1 expression or other markers of immune evasion, despite a significant correlation with higher TIL infiltration (*p* = 0.011) (Table 2A).

Altogether, ALDH1^hi^ alone and in combination with CD44^hi^/CD24^lo^ or Ep-CAM^hi^ correlated with a higher proliferation rate, while it showed no consistent correlation with markers of immune evasion.

CD10 was mainly expressed by the myoepithelial/basal cells of normal-like breast ducts and stroma/mesenchymal cells in some breast cancer tissues. CD10 was overexpressed in cancer cells in 15% of breast cancer cases (Figure 1C). CD10 overexpression did not correlate with PD-L1 expression, other markers of immune evasion (Appendix A), or any other clinicopathological parameter (Appendix A). On the other hand, BMI1 overexpression (27% of cases, Figure 1C) showed paradoxically significant inverse correlation with FOXP3+ CD3+ TIL (*p* = 0.008), loss of p27 (*p* = 0.002), and the combined loss of p21/p27 expression (*p* = 0.006) (Appendix A), tumor size (*p* = 0.003), and ER negative status (*p* = 0.049) (Appendix A). Altogether, CD10 showed no correlation with any of the tested markers, while BMI1 overexpression inversely correlated with some markers of immune evasion and cell proliferation.

### 2.6. The CD44^hi^/CD24^lo^-ALDH^hi^ Combination Correlated with Shorter DFS and OS in Neoadjuvant Chemotherapy-Receiving Patients

We then tested the correlation between different CSC markers and patient outcomes by assessing disease-free survival (DFS) and overall survival (OS). Among all tested CSC markers, CD44^hi^/CD24^lo^-ALDH^hi^ and CD24^hi^ correlated significantly with shorter DFS (*p* = 0.019 and *p* = 0.010, respectively) (Figure 2A). On the other hand, CD44^hi^/CD24^lo^, Ep-CAM^hi^, and ALDH^hi^ CSC markers significantly correlated with shorter OS (*p* = 0.033, *p* = 0.028, and *p* = 0.005, respectively). Meanwhile, the correlation significance of Ep-CAM^hi^/ALDH^hi^ with OS was borderline (*p* = 0.051) (Figure 2B). Remarkably, the CD44^hi^/CD24^lo^-ALDH^hi^ combination marker correlated significantly with shorter OS (*p* < 0.001). Paradoxically, BMI1 did not behave as a CSC marker as its overexpression showed significant inverse correlation with DFS and OS (*p* = 0.037 and *p* = 0.022, respectively).

As the CD44^hi^/CD24^lo^-ALDH^hi^ combination significantly correlated with DFS and OS, we performed a subgroup analysis to determine in which subpopulations this CD44^hi^/CD24^lo^-ALDH^hi^ combination would show correlation with survival. Data stratification analysis revealed a highly significant correlation between CD44^hi^/CD24^lo^-ALDH1^hi^ tumors and DFS and OS (*p* < 0.001) in patients receiving neoadjuvant chemotherapy. On the other hand, in neoadjuvant chemo-naïve patients, CD44^hi^/CD24^lo^-ALDH1^hi^ significantly correlated with OS (*p* = 0.016) but not with DFS (Figure 3A).

Further stratification of patients receiving specific types of neoadjuvant chemotherapy revealed that CD44^hi^/CD24^lo^-ALDH^hi^ tumors were significantly associated with DFS (*p* = 0.001) and OS (*p* < 0.001) among patients treated with Adriamycin/cyclophosphamide (AC; *n* = 37), and also showed significant associations with DFS (*p* = 0.025) and OS (*p* < 0.001) among those treated with fluorouracil/Adriamycin/cyclophosphamide (FAC; *n* = 21) (Appendix A).

To understand the contribution of each CSC marker component, subgroup analysis was performed for ALDH^hi^ and CD44^hi^/CD24^lo^, separately, revealing that ALDH1^hi^ tumors significantly correlated with OS (*p* < 0.001) and DFS (*p* = 0.012) in patients receiving neoadjuvant chemotherapy, but not in neoadjuvant chemotherapy-naïve patients (Appendix A). Similarly, CD44^hi^/CD24^lo^ significantly correlated with OS (*p* = 0.017) in neoadjuvant chemo patients and not chemo-naïve patients (Appendix A). However, the separation in Kaplan–Meier curves was more significant in CD44^hi^/CD24^lo^-ALDH^hi^ tumors than in ALDH^hi^ tumors or CD44^hi^/CD24^lo^ tumors.

Further subgrouping into breast cancer subtypes within neoadjuvant chemotherapy-receiving patients demonstrated that CD44^hi^/CD24^lo^-ALDH^hi^ significantly correlated with DFS and OS in both triple-negative breast cancer (TNBC) patients (*p* = 0.006 and *p* < 0.001, respectively) and non-TNBC patients (*p* < 0.001 for both DFS and OS) (Figure 3B).

Altogether, only the CD44^hi^/CD24^lo^-ALDH^hi^ CSC combination marker correlated significantly with shorter DFS and OS, which was more pronounced in patients who received neoadjuvant chemotherapy.

### 2.7. CD44^hi^/CD24^lo^-ALDH1^hi^ Is an Independent Prognostic Factor for Worse Survival

In univariate analysis, breast cancer cases with CD24^hi^ or CD44^hi^/CD24^lo^-ALDH1^hi^ positive phenotype significantly correlated with DFS (*p* = 0.014 and *p* = 0.022, respectively) (Table 3). In multivariate analysis, CD44^hi^/CD24^lo^-ALDH1^hi^ showed a significant correlation with DFS (*p* = 0.014), as well as CD24^hi^ (*p* = 0.023).

Since several CSC markers showed significant correlation with survival using Kaplan–Meier survival analysis, we conducted a Cox proportional hazards analysis to further identify CSC markers that can serve as independent prognostic factors. Initially, univariate Cox regression of individual CSC markers and their combinations revealed that CD44^hi^/CD24^lo^, Ep-CAM^hi^, ALDH1^hi^, and the CD44^hi^/CD24^lo^-ALDH1^hi^ combination were significantly associated with shorter OS (Table 3). However, BMI1 showed an inverse correlation with OS (*p* = 0.027), which was inconsistent with its role as a CSC marker, leading to its exclusion from further analysis. In multivariate analysis, we included CSC markers that showed significant correlation with survival in our univariate analysis, except CD44^hi^/CD24^lo^-ALDH1^hi^ (excluded to avoid multicollinearity with its individual components). CD44^hi^/CD24^lo^ (*p* = 0.039*)* and ALDH1^hi^ (*p* = 0.009) showed significant correlations with OS, whereas Ep-CAM^hi^ lost significance. Rerunning the multivariate analysis while including the CD44^hi^/CD24^lo^-ALDH1^hi^ combination CSC marker and avoiding its individual components (ALDH^hi^ and CD44^hi^/CD24^lo^) demonstrated a significant correlation with OS (*p* < 0.001) while Ep-CAM^hi^ lost significance (Table 3). Altogether, CD44^hi^/CD24^lo^-ALDH1^hi^ was an independent prognostic factor for worse DFS and OS, as shown by univariate and multivariate Cox regression hazards analysis.

In summary, the vast majority of CSC markers and their combinations were significantly associated with the expression of proliferation markers (Table 4), with the exception of CD10, while BMI1 correlated inversely with proliferation. Similarly, almost all CSC phenotypes correlated with ER-negative status. On the other hand, significant correlations were observed with EMT and immune evasion markers for CD44^hi^/CD24^lo^, Ep-CAM^hi^, the CD44^hi^/CD24^lo^-Ep-CAM^hi^ combination, and CD24^hi^, but not for ALDH1^hi^, the combinations CD44^hi^/CD24^lo^-ALDH1^hi^, or Ep-CAM^hi^/ALDH1^hi^. Overall, the CD44^hi^/CD24^lo^-ALDH1^hi^ combination CSC marker was an independent prognostic factor for survival, which was more significant in patients receiving neoadjuvant chemotherapy.

## 3. Discussion

Cancer stem cells (CSCs) can exist in different types or states within the same tumor, which can be identified by specific phenotypic or functional markers [10]. Both intrinsic and extrinsic factors drive the capacity of CSCs to self-renew, proliferate, and metastasize [15]. The state/type that is associated with a specific cancer trait has not been well established. In the current study, we investigated the association between various CSC markers and key oncogenic traits in breast cancer. This report shows that the most common feature of all tested CSC markers was their association with higher proliferation and estrogen receptor negativity. Importantly, we have demonstrated for the first time that breast tumors with CD24^hi^ or Ep-CAM^hi^ CSCs were associated with markers of immune evasion, while CD44^hi^/CD24^lo^ and Ep-CAM^hi^ CSCs were associated with features of EMT. On the other hand, the combination of CD44^hi^/CD24^lo^-ALDH1^hi^ correlated with worse survival, especially in patients receiving neoadjuvant chemotherapy.

We have previously shown, using multi-parametric flow cytometry, that the expression of PD-L1 is higher in breast CSC-like cells with the CD44^hi^/CD24^lo^ phenotype, but not in ALDH^hi^ cells. Whether this could be validated in breast cancer patients remained an intriguing question. In this report, we demonstrated, using breast cancer patient samples, that PD-L1 expression significantly correlates with CD44^hi^/CD24^lo^ but not with ALDH1^hi^. The results in this report align with the previously reported correlation between PD-L1 and CD44^hi^/CD24^lo^ [16], but contradict the earlier reported correlation of PD-L1 with ALDH1 by Flores et al. [17]. This discrepancy could be due to the variation in the anti-ALDH1 antibody used. We utilized the ALDH1 antibody (clone 44/ALDH), while Flores et al. employed an antibody from Abcam (the clone was not specified). Additionally, we report for the first time that PD-L1 expression in breast cancer correlates with Ep-CAM^hi^, another marker for CSCs, and the combination of CD44^hi^/CD24^lo^-Ep-CAM^hi^, as well as CD24^hi^. Specifically, Ep-CAM^hi^ and CD24^hi^ showed association with multiple markers of immune evasion.

Our findings were consistent with previous data showing a correlation between CD44^hi^/CD24^lo^ phenotype and high histological grade [18], ER negativity status [18,19,20], Ki-67 positivity status [20,21], and vimentin upregulation [22]. In the process of EMT, epithelial cells upregulate CD44 and downregulate CD24 [23]. Indeed, in the current study, breast cancer with CD44^hi^/CD24^lo^ cells showed a significant correlation with standard markers of EMT, specifically the upregulation of vimentin and downregulation of E-cadherin, which is consistent with previous reports on the relationship between EMT and CSCs [23,24,25]. On the other hand, the effect of CD44^hi^/CD24^lo^ on survival has been inconsistent in previous studies. While a few reports have demonstrated a correlation with survival [19,26], the majority did not [27]. In this report, we have shown a correlation with OS but not with DFS. The discrepancy in the results of CD44^hi^/CD24^lo^ previously reported in breast cancer was possibly due to differences in the (1) scoring method, (2) the CD24 antibody used especially clone SN3b, which is not very specific [28], (3) the intensity of CD44 or CD24 that was considered positive, and (4) whether the cohort of breast cancer patients used have received neoadjuvant chemotherapy therapy or not. In the current study, we have used a very well-established CD24 antibody (clone ML5) [28]. Scorings of CD24 were considered low/negative (CD24^lo^) if they were totally negative or below the intensity of normal epithelial ducts, which is considered +1 intensity of CD24, and neoadjuvant patients were analyzed separately, demonstrating that the correlation of CD44^hi^/CD24^lo^ was with patients who received neoadjuvant chemotherapy (Appendix A).

In this report, overexpression of Ep-CAM (Ep-CAM^hi^) was associated with higher tumor size, hormone receptor negativity status, and Ki-67 positivity status, which is consistent with previous reports that used the exact antibody clone (VU-1D9) and scoring method (H-score) [29,30,31]. We have further shown for the first time the correlation of Ep-CAM^hi^ with the loss of p27 and the loss of both p21/p27 in breast cancer, consistent with the correlation of Ep-CAM with a higher proliferation rate. We have shown in this report for the first time the correlation of Ep-CAM^hi^ with makers of immune evasion, namely higher expression of PD-L1, higher infiltration with FOXP3 + TIL (T-reg), and PD-1+ TIL. Indeed, Chen et al. demonstrated the role of Ep-CAM signaling in promoting PD-L1 expression and the significant correlation of Ep-CAM expression with PD-L1 expression in lung cancer [32]. Furthermore, Zheng et al. have shown that Ep-CAM, as a tumor-associated antigen, promoted a Th2 type of immune response and tumor immune evasion [33]. Ep-CAM overexpression significantly correlated with worse OS, which was consistent with previously reported data [30,31,34,35].

CD24 has an established role in immune evasion through the binding to Siglec-10 in macrophages [36] and natural killer cells [37]. To our knowledge, this is the first study to report on the correlation of CD24 expression with immune evasion in breast cancer, including PD-L1 expression, FOXP3 + TIL, and PD-1 + TIL. Further studies are warranted to investigate whether such interaction creates an appropriate microenvironment for the accumulation of immune suppressive lymphocytes like FOXP3 + TIL and PD-1 + TIL, and whether CD24 expression can functionally affect the clinical outcomes.

The fact that CD24 is an important driver of cell proliferation in many types of cancer [38], including breast cancer [39,40], is consistent with its correlation with Ki-67, SKP2 expression, and the lack of p21 and p27 cell cycle checkpoints expression that we observed in our breast cancer samples. In addition, CD24 was reported to be downregulated by estrogen [41], which is consistent with the correlation between CD24 overexpression and ER negativity status we observed in this report.

While previous studies have shown a significant association between ALDH1 expression and overall survival, the data in the literature have not been consistent on this correlation. Our finding that ALDH1 was highly associated with both DFS and OS only in neoadjuvant chemotherapy-receiving patients (Appendix A) is consistent with previous reports [42,43]. Importantly, this is the first report to demonstrate a highly significant correlation between CD44^hi^/CD24^lo^-ALDH1^hi^ and survival (OS and DFS) in patients receiving neoadjuvant chemotherapy. The fact that the tissues were surgical samples suggests that CSCs with the CD44^hi^/CD24^lo^-ALDH1^hi^ phenotypes are resistant to chemotherapy, as their existence had the highest prognostication ability among all other tested CSC markers.

Our findings of no correlation between ALDH1 and other immune evasion markers, such as FOXP3 + TIL and PD-1 + TIL, disagree with those of Seo et al. [44], which demonstrated a significant correlation between ALDH1 and FOXP3 + TIL. This discrepancy is most likely due to the scoring method Seo et al. used, which only considered high-intensity ALDH1 and a 10% cutoff. Indeed, if we used their scoring method, we would obtain a significant correlation with FOXP3 + TIL but not with PD-L1 or PD-1 + TIL in our dataset. Similar to Seo et al., this scoring method will give a positivity rate of 10% in our cohort of breast cancer patients, but this percentage is far from the average positivity of 20–50% seen in most of the 15 studies related to ALDH1 in breast cancer that were reviewed by Liu et al. [45]. In contrast to our study, Seo et al. [44] did not study PD-1 + TIL or PD-L1 in cancer cells. It is worth mentioning in this regard that several studies have demonstrated a correlation between ALDH1 expression and other immune evasion markers, such as the myeloid suppressor-derived cells (MSDC) [46], which was not studied in this report.

CD10 did not correlate with any clinicopathological parameters, although it showed a trend of correlation with ER and PR negativity status, as previously reported [47]. A larger cohort of breast cancer patients may be needed, as only a small percentage (15%; *n* = 11) of our breast cancer patients showed CD10 positivity.

BMI1 is a component of the polycomb group complex 1, an essential epigenetic repressor of multiple regulatory genes and a driver for breast CSCs [8,48]. The literature has been inconsistent on the prognostic role of BMI1 in breast cancer. Choi et al. [49] have shown that BMI1 is a good prognostic factor, which is consistent with the results in the current report. Conversely, Wang et al. [50] have shown that BMI1 is a bad prognostic factor in breast cancer. Althobiti et al. [51] attempted to reconcile the difference by demonstrating that the prognostic effect of BMI1 is breast cancer subtype-dependent, as it is a favorable prognostic factor in luminal breast cancer and an unfavorable prognostic factor in TNBC. However, even if the patients in this report were segregated into different subtypes, the trend is still towards good prognosis in all subtypes, although the significance would be lost due to the small number in each subtype.

Previously, it was shown that patients with multiple CSC markers have worse survival [21]. However, which combination has the most significant correlation with survival was not addressed. In our study, we have shown that only the CD44^hi^/CD24^lo^ and ALDH1 combination significantly correlates with DFS and OS, especially in neoadjuvant chemotherapy-receiving breast cancer patients, and this was observed in both TNBC and non-TNBC subtypes.

One limitation of our study is that it relies on a small cohort of breast cancer patients who were treated between 2004 and 2010, an era before immunotherapy and before neoadjuvant anti-HER2 therapy, which might affect the DFS and OS. Therefore, findings in this report should be confirmed in an independent recent cohort of breast cancer patients. On the other hand, the advantage of this cohort is the long-term survival data and the accumulated data we had from previous studies. Another limitation of this study is that we have studied breast cancer as a whole, although we tried our best to mitigate this by subgroup analysis; however, subtype-focused studies should be carried out in the future to confirm these findings.

## 4. Materials and Methods

### 4.1. Patients

This study was conducted under the Helsinki Declaration and approved by the Institutional Review Board of King Faisal Specialist Hospital and Research Centre (KFSH&RC). The archived FFPE tissues were surgical tumor tissues excised as lumpectomy or mastectomy from patients diagnosed with invasive ductal carcinoma (IDC), as previously described [52,53]. All patients provided informed consent.

### 4.2. Immunohistochemistry

Sections (4 microns) of formalin-fixed paraffin-embedded (FFPE) breast cancer tissue blocks were dewaxed in xylene and rehydrated with gradients of alcohol and water. Antigen retrieval took place in the Decloaking Chamber pressure cooker (Biocare, Pacheco, CA, USA), as detailed in Table 5. Endogenous peroxidase was quenched using 0.9% H_2_O_2_, while endogenous biotin was blocked with biotin and avidin, along with three washes in between. Primary antibodies, diluted in 1% BSA, were incubated overnight at 4 °C. For the CD44/CD24 analysis, the two-species double-staining method was employed (CD44 is a rabbit antibody, while CD24 is a mouse antibody). A biotinylated streptavidin Alkaline Phosphatase (AP) anti-mouse secondary antibody was utilized, followed by Envision anti-rabbit horseradish peroxidase (HRP) (Agilent, Dako, Glostrup, Denmark). In the case of single staining for Ep-CAM, ALDH1, and BMI1, the secondary antibody used was either Envision anti-mouse or a biotinylated antibody (Jackson Laboratories, Bar Harbor, ME, USA) (Table 5). Color development was accomplished using Fast Red (30 min) and/or DAB (10 min). All washes were conducted using TBST. CD10, PD-L1, and Vimentin immunohistochemistry were performed using the fully automated Ventana Benchmark Ultra system with Ventana primary antibodies, as previously described in detail [52,54]. Data on Vimentin/E-cadherin expression, as well as SKP2/p21/p27, were obtained from two previously published studies on the same patient cohort [52,55]. On the other hand, FOXP3 and PD-1 data were derived from prior work conducted on frozen tissue sections from the same cohort [56].

Scoring of CSCs was performed by an anatomical pathologist (AT) using 5–10% increments. Borderline CD44^hi^ (i.e., 10%) was considered positive only if the intensity was ≥2+. Due to high heterogeneity, we used the H-score system to score ALDH1, Ep-CAM, and CD24 expression in cancer cells. The data were further dichotomized using 35, 200, and 140 as cutoffs for ALDH1, Ep-CAM, and CD24, respectively. All cutoffs were determined using the receiver operating characteristic (ROC) curve/Youden index.

### 4.3. Statistical Analysis

The correlation between CSC markers and hallmarks of cancer (categorical variables) was evaluated using Fisher’s exact test. Survival plots were generated using the Kaplan–Meier method, and the curves were compared using the log-rank test. The univariate and multivariate Cox proportional hazards models were used to test the correlation with survival. Time was censored for patients who were alive or disease-free at the last follow-up, including patients lost to follow-up. All analyses, including the ROC curve/Youden index analyses, were performed using JMP statistical software (version 15; SAS Institute, Cary, NC, USA), and a *p*-value of 0.05 was set as the threshold for significance.

## 5. Conclusions

Multiple markers have been identified for breast cancer stem cells (CSCs), reflecting the heterogeneity of their phenotypes and states. Despite their differences, CSCs defined by these markers share common oncogenic characteristics, such as their association with estrogen receptor (ER) negativity and markers of higher proliferative rate. Certain phenotypes—including CD44^hi^/CD24^lo^, EpCAM^hi^, and CD24^hi^, and the CD44^hi^/CD24^lo^-EpCAM^hi^ combination—are also associated with features of epithelial-to-mesenchymal transition (EMT) and immune evasion. Moreover, certain CSC markers, such as CD44^hi^/CD24^lo^, EpCAM^hi^, and ALDH1^hi^, have been linked to poorer overall survival. Notably, the CD44^hi^/CD24^lo^-ALDH1^hi^ combination emerged as the most significant independent prognostic factor for survival, particularly in patients who received neoadjuvant chemotherapy. The choice of CSC marker depends on the features/hallmarks features or hallmarks under investigation. In the future, distinct CSC markers can be leveraged to trace/monitor various disease characteristics or treatment outcomes.

## Figures and Tables

**Figure 1 ijms-26-08219-f001:**
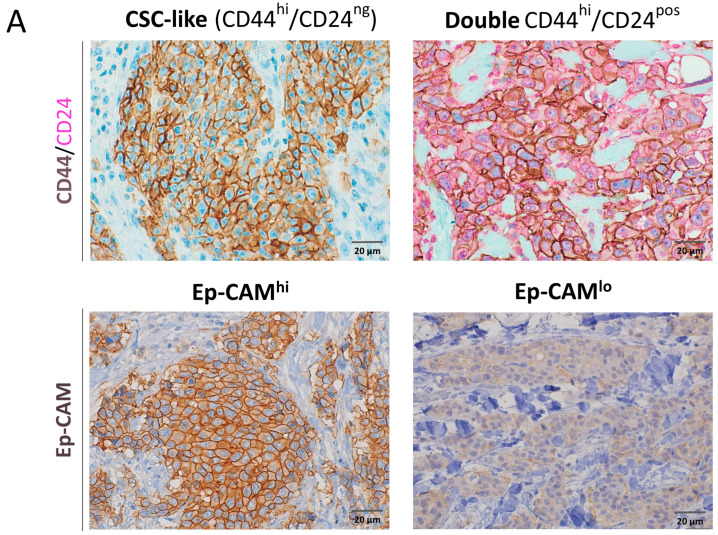
Expression of cancer stem cell (CSC) markers in breast cancer tissue sections. Representative immunohistochemical images (magnification ×400) showing the expression of CSC-related markers in breast cancer tissues. (**A**) Dual staining for CD44 and CD24, showing membranous CD44 (brown) and cytoplasmic CD24 (purple) (top) and membranous Ep-CAM (bottom). (**B**) Cytoplasmic CD24 and ALDH1; note the positivity in stromal cells in the ALDH1lo image. (**C**) Membranous CD10 and nuclear BMI1.

**Figure 2 ijms-26-08219-f002:**
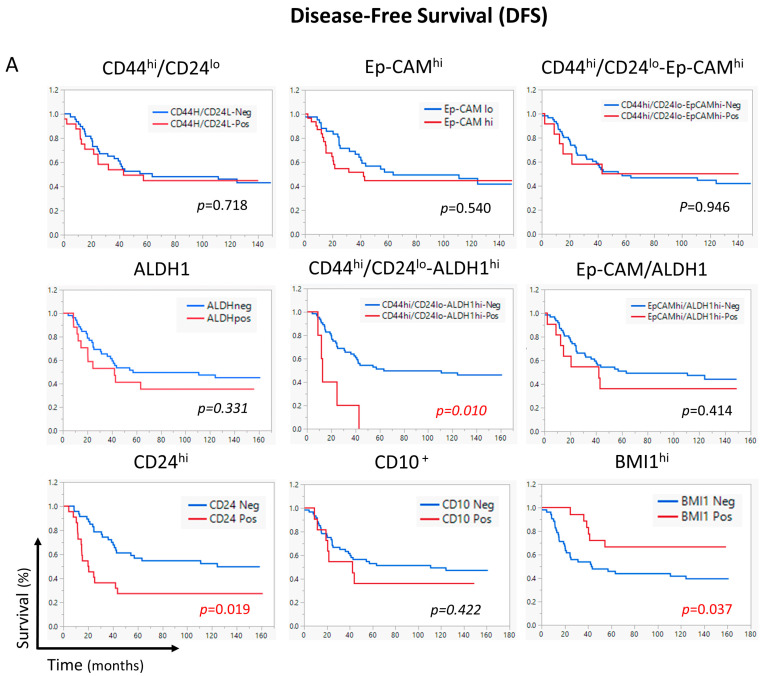
Correlation of CSCs expression with the Survival of breast cancer patients. Kaplan-Meier survival curves showing the association between CSCs and disease-free survival (DFS) (**A**) or overall survival (OS) (**B**) of breast cancer patients. Statistical significance was determined using the log-rank test.

**Figure 3 ijms-26-08219-f003:**
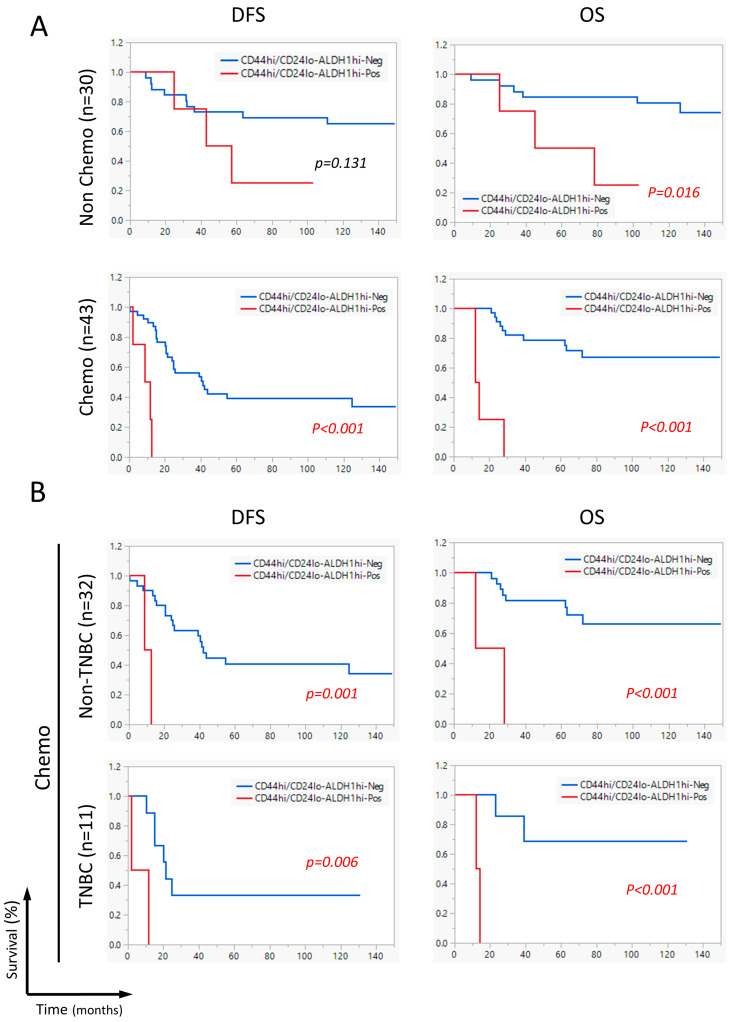
Subgroup analysis in neoadjuvant chemotherapy-receiving patients and TNBC patients. Kaplan–Meier survival curves illustrate the association between CD44^hi^/CD24^lo**-**^ALDH^hi^ breast CSCs and disease-free survival (DFS) or overall survival (OS), depending on neoadjuvant chemotherapy status (**A**) and breast cancer subtype in neoadjuvant chemotherapy receiving patients (**B**). Statistical significance was determined using the log-rank test.

**Table 1 ijms-26-08219-t001:** Correlation between CD44^hi^/CD24^lo^ or Ep-CAM^hi^ CSCs and immune /proliferation markers (**A**) or other clinicopathological parameters (**B**) in 73 breast cancer patients.

**(A)**										
		**CD44^hi^/CD24^lo^**			**Ep-CAM^hi^**			**CD44^hi^/CD24^lo^-** **Ep-CAM^hi^**		
		**<10%**	**≥10%**	**^♣^ *p***	**H-Score < 200**	**H-Score ≥ 200**	**^♣^ *p***	**−**	**+**	**^♣^ *p***
Immune Markers	**PD-L1**									
	<5%	**45 (76)** *	**14 (24)**	**0.001**	**38 (64)**	**21 (36)**	**0.019**	**53 (90)**	**6 (10)**	**0.008**
	≥5%	**4 (29)**	**10 (71)**		**4 (29)**	**10 (71)**		**8 (57)**	**6 (43)**	
	**TIL**									
	Low (Score 1 and 2)	40 (70)	17 (30)	0.369	**37 (65)**	**20 (35)**	**0.024**	50 (88)	7 (12)	0.120
	High (Score 3)	9 (56)	7 (44)		**5 (31)**	**11 (69)**		11 (69)	5 (31)	
	^§^ **FOXP3 + TIL**									
	<10%	27 (73)	10 (27)	0.784	**26 (70)**	**11 (30)**	**0.012**	33 (89)	4 (11)	0.483
	≥10%	19 (68)	9 (32)		**11 (39)**	**17 (61)**		23 (82)	5 (18)	
	^§^ **PD-1 + TIL**									
	<10%	24 (75)	8 (25)	0.314	23 (72)	9 (28)	0.052	30 (94)	2 (6)	0.097
	≥10%	24 (63)	14 (37)		18 (47)	20 (53)		30 (79)	8 (21)	
Proliferation Markers	**Ki-67**									
	<20%	**29 (81)**	**7 (19)**	**0.024**	**26 (72)**	**10 (28)**	**0.018**	**34 (94)**	**2 (6)**	**0.024**
	≥20%	**20 (54)**	**17 (46)**		**16 (43)**	**21 (57)**		**27 (73)**	**10 (27)**	
	^§^ **SKP2**									
	<10%	35 (76)	11 (24)	0.062	29 (63)	17 (37)	0.314	40 (87)	6 (13)	0.322
	≥10%	13 (52)	12 (48)		12 (48)	13 (52)		19 (76)	6 (24)	
	^§^ **p21**									
	<10%	28 (65)	15 (35)	0.615	22 (51)	21 (49)	0.221	34 (79)	9 (21)	0.341
	≥10%	20 (71)	8 (29)		19 (68)	9 (32)		25 (89)	3 (11)	
	^§^ **p27**									
	<50%	33 (65)	18 (35)	0.574	**25 (49)**	**26 (51)**	**0.031**	41 (80)	10 (20)	0.488
	≥50%	15 (75)	5 (25)		**16 (80)**	**4 (20)**		18 (90)	2 (10)	
	^§^ **Loss of both p21/p27**									
	NO	17 (55)	14 (45)	0.072	**28 (70)**	**12 (30)**	**0.029**	23 (74)	8 (26)	0.112
	YES	31 (78)	9 (23)		**13 (42)**	**18 (58)**		36 (90)	4 (10)	
(**B**)										
		**CD44^hi^ CD24^lo^**			**Ep-CAM^hi^**			**CD44^hi^ CD24^lo^/** **Ep-CAM^hi^**		
		**<10%**	**≥10%**	**^♣^ *p***	**H-Score < 200**	**H-Score ≥ 200**	**^♣^ *p***	**−**	**+**	**^♣^ *p***
Clinicopathological Markers	**Age**									
	<40 years	14 (58) *	10 (42)	0.297	13 (54)	11 (46)	0.802	17 (71)	7 (29)	0.051
	≥40 years	35 (71)	14 (29)		29 (59)	20 (41)		44 (90)	5 (10)	
	**Tumor Size**									
	<4 cm	30 (77)	9 (23)	0.081	**27 (69)**	**12 (31)**	**0.036**	34 (87)	5 (13)	0.529
	≥4 cm	19 (56)	15 (44)		**15 (44)**	**19 (56)**		27 (79)	7 (21)	
	**Invasion**									
	Absent	21 (70)	9 (30)	0.801	16 (53)	14 (47)	0.633	25 (83)	5 (17)	1.000
	Present	28 (65)	15 (35)		26 (60)	17 (40)		36 (84)	7 (16)	
	**Histological Grade**									
	1 and 2	**30 (81)**	**7 (19)**	**0.013**	25 (68)	12 (32)	0.100	34 (92)	3 (8)	0.064
	3	**19 (53)**	**17 (47)**		17 (47)	19 (53)		27 (75)	9 (25)	
	**^§^ Lymph Node Metastasis**									
	Absent	18 (69)	8 (31)	1.000	14 (54)	12 (46)	0.810	20 (77)	6 (23)	0.331
	Present	31 (67)	15 (33)		27 (59)	19 (41)		40 (87)	6 (13)	
	**Neoadjuvant Chemotherapy**									
	Absent	21 (70)	9 (30)	0.801	19 (63)	11 (37)	0.476	27 (90)	3 (10)	0.337
	Present	28 (65)	15 (35)		23 (54)	20 (47)		34 (79)	9 (21)	
BC Subtype Markers	**ER Status**									
	Negative	**12 (44)**	**15 (56)**	**0.002**	**10 (37)**	**17 (63)**	**0.008**	**18 (67)**	**9 (33)**	**0.007**
	Positive	**37 (80)**	**9 (20)**		**32 (70)**	**14 (30)**		**43 (93)**	**3 (7)**	
	**PR Status**									
	Negative	23 (58)	17 (42)	0.080	**17 (42)**	**23 (58)**	**0.005**	**29 (72)**	**11 (28)**	**0.009**
	Positive	26 (79)	7 (21)		**25 (76)**	**8 (24)**		**32 (97)**	**1 (3)**	
	**HER2/neu Status**									
	Negative	33 (67)	16 (33)	1.000	29 (59)	20 (41)	0.802	40 (82)	9 (18)	0.739
	Positive	16 (67)	8 (33)		13 (54)	11 (46)		21 (88)	3 (12)	
	**TNBC Status**									
	Negative	42 (72)	16 (28)	0.071	**38 (66)**	**20 (34)**	**0.009**	**53 (91)**	**5 (9)**	**0.002**
	Positive	7 (47)	8 (53)		**4 (27)**	**11 (73)**		**8 (53)**	**7 (47)**	
EMT Markers	**Vimentin**									
	Negative	**42 (82)**	**9 (18)**	**<0.001**	33 (65)	18 (35)	0.074	**47 (92)**	**4 (8)**	**0.005**
	Positive	**7 (32)**	**15 (68)**		9 (41)	13 (59)		**14 (64)**	**8 (36)**	
	**Loss of E-Cadherin**									
	Negative	**39 (76)**	**12 (24)**	**0.015**	32 (63)	19 (37)	0.203	**47 (92)**	**4 (8)**	**0.005**
	Positive	**10 (45)**	**12 (55)**		10 (45)	12 (55)		**14 (64)**	**8 (36)**	
	**Vimentin/Loss of E-Cadherin**									
	Negative	**47 (76)**	**15 (24)**	**<0.001**	**39 (63)**	**23 (37)**	**0.045**	**56 (90)**	**6 (10)**	**0.002**
	Positive	**2 (18)**	**9 (82)**		**3 (27)**	**8 (73)**		**5 (45)**	**6 (55)**	

Abbreviations: * the numbers between brackets are the percentages of patients and ^♣^
*p* values in bold and shaded represent significant data. Light shading represents borderline significance. ^§^ PD-1 is unknown for 3 samples, FOXP3 is unknown for 6 samples. The status is unknown for SKP2, p21, p27, and loss of both p21/p27 status for 2 cases. The lymph node status is missing for one case.

**Table 2 ijms-26-08219-t002:** Correlation between ALDH1^hi^ or other CSCs and immune /proliferation markers (**A**) or other clinicopathological parameters (**B**) in 73 breast cancer patients.

(**A**)										
		**ALDH1^hi^**			**CD44^hi^/CD24^lo^-ALDH1^hi^**			**Ep-CAM^hi^/ALDH1^hi^**		
		**H-Score < 35**	**H-Score ≥ 35**	**^♣^ *p***	**−**	**+**	**^♣^ *p***	**−**	**+**	**^♣^ *p***
Immune Markers	**PD-L1**									
	<5%	41 (69) *	18 (31)	1.000	54 (92)	5 (8)	0.174	51 (86)	8 (14)	0.432
	≥5%	10 (71)	4 (29)		11 (79)	3 (21)		11 (79)	3 (21)	
	**TIL**									
	Low (Score 1 and 2)	**44 (77)**	**13 (23)**	**0.015**	53 (93)	4 (7)	0.064	**52 (91)**	**5 (9)**	**0.011**
	High (Score 3)	**7 (44)**	**9 (56)**		12 (75)	4 (25)		**10 (62)**	**6 (38)**	
	^§^ **FOXP3 + TIL**									
	<10%	29 (78)	8 (22)	0.170	34 (92)	3 (8)	1.000	33 (89)	4 (11)	0.483
	≥10%	17 (61)	11 (39)		25 (89)	3 (11)		23 (82)	5 (18)	
	^§^ **PD-1 + TIL**									
	<10%	24 (75)	8 (25)	0.603	30 (94)	2 (6)	0.681	29 (91)	3 (9)	0.494
	≥10%	26 (68)	12 (32)		34 (89)	4 (11)		32 (84)	6 (16)	
Proliferation Markers	**Ki-67**									
	<20%	29 (81)	7 (19)	0.074	35 (97)	1 (3)	0.056	**34 (94)**	**2 (6)**	**0.046**
	≥20%	22 (59)	15 (41)		30 (81)	7 (19)		**28 (80)**	**9 (20)**	
	^§^ **SKP2**									
	<10%	34 (74)	12 (26)	0.423	42 (91)	4 (9)	0.440	41 (89)	5 (11)	0.307
	≥10%	16 (64)	9 (36)		21 (84)	4 (16)		20 (80)	5 (20)	
	^§^ **p21**									
	<10%	29 (67)	14 (33)	0.599	36 (84)	7 (16)	0.136	34 (79)	9 (21)	0.077
	≥10%	21 (75)	7 (25)		27 (96)	1 (4)		27 (96)	1 (4)	
	^§^ **p27**									
	<50%	**32 (63)**	**19 (37)**	**0.041**	43 (84)	8 (16)	0.095	41 (80)	10 (20)	0.053
	≥50%	**18 (90)**	**2 (10)**		20 (100)	0 (0)		20(100)	0 (0)	
	**Loss of both p21/p27**									
	NO	19 (61)	12 (39)	0.191	**24 (77)**	**7 (23)**	**0.018**	**22 (71)**	**9 (29)**	**0.002**
	YES	31 (78)	9 (22)		**39 (98)**	**1 (2)**		**39 (98)**	**1 (3)**	
(**B**)										
		**ALDH1^hi^**			**CD44^hi^/CD24^lo^-ALDH1^hi^**			**ALDH1^hi^/Ep-CAM^hi^**		
		**H-Score < 35**	**H-Score ≥ 35**	**^♣^ *p***	**−**	**+**	**^♣^ *p***	**−**	**+**	**^♣^ *p***
Clinicopathological Markers	**Age**									
	<40 years	19 (79) *	5 (21)	0.284	21 (88)	3 (12)	1.000	20 (83)	4 (17)	1.000
	≥40 years	32 (65)	17 (35)		44 (90)	5 (10)		42 (86)	7 (14)	
	**Tumor Size**									
	<4 cm	**34 (87)**	**5 (13)**	**<0.001**	**38 (97)**	**1 (3)**	**0.022**	**37 (95)**	**2 (5)**	**0.019**
	≥4 cm	**17 (50)**	**17 (50)**		**27 (79)**	**7 (21)**		**25 (74)**	**9 (26)**	
	**Invasion**									
	Absent	22 (73)	8 (27)	0.616	28 (93)	2 (7)	0.458	26 (87)	4 (13)	1.000
	Present	29 (67)	14 (33)		37 (86)	6 (14)		36 (84)	7 (16)	
	**Histological Grade**									
	1 and 2	**31 (84)**	**6 (16)**	**0.011**	**36 (97)**	**1 (3)**	**0.028**	34 (92)	3 (8)	0.112
	3	**20 (56)**	**16 (44)**		**29 (81)**	**7 (19)**		28 (78)	8 (22)	
	^§^ **Lymph Node Metastasis**									
	Absent	19 (73)	7 (27)	0.794	25 (96)	1 (4)	0.410	24 (92)	2 (8)	0.307
	Present	32 (70)	14 (30)		40 (87)	6 (13)		37 (80)	9 (20)	
	**Neoadjuvant Chemotherapy**									
	Absent	17 (57)	13 (43)	0.068	26 (87)	4 (13)	0.710	25 (83)	5 (17)	0.752
	Present	34 (79)	9 (21)		39 (91)	4 (9)		37 (86)	6 (14)	
BC Subtype Markers	**ER Status**									
	Negative	16 (59)	11 (41)	0.187	**21 (78)**	**6 (22)**	**0.045**	**19 (70)**	**8 (30)**	**0.015**
	Positive	35 (76)	11 (24)		**44 (96)**	**2 (4)**		**43 (93)**	**3 (7)**	
	**PR Status**									
	Negative	26 (65)	14 (35)	0.443	33 (83)	7 (17)	0.065	31 (78)	9 (23)	0.097
	Positive	25 (76)	8 (24)		32 (97)	1 (3)		31 (94)	2 (6)	
	**HER2/neu Status**									
	Negative	37 (76)	12 (24)	0.176	44 (90)	5 (10)	1.000	43 (88)	6 (12)	0.487
	Positive	14 (58)	10 (42)		21 (88)	3 (12)		19 (79)	5 (21)	
	**TNBC Status**									
	Negative	41 (71)	17 (29)	0.760	53 (91)	5 (9)	0.348	51 (88)	7 (12)	0.221
	Positive	10 (67)	5 (33)		12 (80)	3 (20)		11 (73)	4 (27)	
EMT Markers	**Vimentin**									
	Negative	35 (69)	16 (31)	0.788	**48 (94)**	**3 (6)**	**0.049**	44 (86)	7 (14)	0.724
	Positive	16 (73)	6 (27)		**17 (77)**	**5 (23)**		18 (82)	4 (18)	
	**Loss of E-Cadherin**									
	Negative	38 (75)	13 (25)	0.266	47 (92)	4 (8)	0.232	46 (90)	5 (10)	0.077
	Positive	13 (59)	9 (41)		18 (82)	4 (18)		16 (72)	6 (27)	
	**Vimentin/Loss** **of E-Cadherin**									
	Negative	44 (71)	18 (29)	0.724	57 (92)	5 (8)	0.095	55 (89)	7 (11)	0.055
	Positive	7 (64)	4 (36)		8 (73)	3 (27)		7 (64)	4 (36)	

Abbreviations: * The numbers between brackets are the percentages of patients and ^♣^
*p* values in bold and shaded represent significant data. Light shading represents borderline significance. ^§^ PD-1 is unknown for 3 samples, FOXP3 is unknown for 6 samples. The status is unknown for SKP2, p21, p27, and loss of both p21/p27 status for 2 cases. The lymph node status for one case is missing.

**Table 3 ijms-26-08219-t003:** Cox proportional hazard regression analysis of clinicopathological features with disease-free survival (DFS) and overall survival (OS) in 73 patients with breast cancer.

	Relapse		RFS						Death		OS								
			Univariate			Multivariate					Univariate			Multivariate			Multivariate		
	−	+	HR	95% CI	* *p*	HR	95%CI	^♣^ *p*	−	+	HR	95% CI	* *p*	HR	95% CI	* *p*			
**CD44^hi^ CD24^lo^**																			
<10%	22 (45) *	27 (55)	1						**38 (78)**	**11 (22)**	**1**			**1**					
≥10%	11 (46)	13 (54)	1.1	0.6–2.2	0.718				**12 (50)**	**12 (50)**	**2.4**	**1.0–5.4**	**0.039**	**2.4**	**1.1–5.6**	**0.039**			
**Ep-CAM^hi^**																			
<H-Score 200	19 (45)	23 (55)	1						**33 (79)**	**9 (21)**	**1**			1			1		
≥H-Score 200	14 (45)	17 (55)	1.2	0.6–2.3	0.541				**17 (55)**	**14 (45)**	**2.5**	**1.1–5.7**	**0.034**	1.9	0.8–4.6	0.131	2.0	0.8–4.7	0.113
**CD44^hi^ CD24^lo^-** **Ep-CAM^hi^**																			
<H-Score 200	27 (44)	34 (56)	1						44 (72)	17 (28)	1								
≥H-Score 200	6 (50)	6 (50)	1.0	0.4–2.3	0.946				6 (50)	6 (50)	2.0	0.8–5.1	0.140						
**ALDH1^hi^**																			
<H-Score 35	24 (47)	27 (53)	1						**40 (78)**	**11 (22)**	**1**			**1**					
≥H-Score 35	9 (41)	13 (59)	1.2	0.6–2.3	0.630				**10 (45)**	**12 (55)**	**3.0**	**1.3–6.9**	**0.008**	**3.0**	**1.3–7.0**	**0.009**			
**CD44^hi^/CD24^lo^-ALDH1^hi^**																			
Negative	**32 (49)**	**33 (51)**	1			1			**49 (75)**	**16 (25)**	**1**						**1**		
Positive	**1 (12)**	**7 (88)**	**2.8**	**1.2–6.4**	**0.014**	**2.8**	**1.2–6.4**	**0.014**	**1 (13)**	**7 (87)**	**6.4**	**2.6–15.8**	**<0.001**				**5.4**	**2.1–13.6**	**<0.001**
**Ep-CAM^hi^/ALDH1^hi^**																			
Negative	29 (47)	33 (53)	1						45 (73)	17 (27)	1								
Positive	4 (36)	7 (64)	1.4	0.6–3.2	0.417				5 (45)	6 (55)	2.5	1.0–6.2	0.059						
**CD24^hi^**																			
<H-Score 140	**25 (52)**	**23 (48)**	**1**			**1**			33 (69)	15 (31)	1								
≥H-Score 140	**8 (32)**	**17 (68)**	**2.1**	**1.1–3.9**	**0.022**	**2.2**	**1.1–4.3**	**0.023**	17 (68)	8 (32)	1.4	0.6–3.3	0.460						
**CD10**																			
<10%	28 (47)	32 (53)	1						44 (73)	16 (27)	1								
≥10%	4 (36)	7 (64)	1.3	0.6–3.0	0.491				5 (45)	6 (55)	2.3	0.9–5.9	0.081						
**BMI1**																			
Negative	**20 (38)**	**32 (62)**	**1**						**32 (62)**	**20 (38)**	**1**								
Positive	**12 (63)**	**7 (37)**	**0.4**	**0.2–1.0**	**0.043**				**17 (89)**	**2 (11)**	**0.2**	**0.05–0.8**	**0.027**						

Abbreviations: * (+ and −) are the number of positive and negative patients, the numbers between brackets are the percentages of patients, and ^♣^
*p* values in bold and shaded areas represent significant data. Light shading represents borderline significance.

**Table 4 ijms-26-08219-t004:** Summarized results of correlation between CSC types/states with hallmarks of cancer and survival of breast cancer patients (*n* = 73).

	CD44/CD24	Ep-CAM	CD44/CD24- Ep-CAM	ALDH1	CD44/CD24-ALDH1	Ep-CAM/ ALDH1	CD24	CD10
EMT	√√√ *	√	√√√		√		√	
Higher Proliferation	√	√√√√	√	√	√	√√	√√√	
Immune Evasion	√	√√√	√				√√√	
Higher Histological Grade	√			√	√		√	
ER negative	√	√√√	√		√	√	√	

Abbreviations: * Number of checks represents the number of markers that demonstrate a significant correlation. Highlighted traits are common in all tested CSC markers except CD10.

**Table 5 ijms-26-08219-t005:** Conditions used for manual Immunohistochemistry on FFPE tissues.

Antibody	Antigen Retrieval		Primary Antibody		Secondary Antibody	
	Solution	Conditions *	Dilution	Clone/Cat#		
**CD44/CD24**	Tris-EDTA pH9	121 °C/7 min	CD24, 1:100 CD44, 1:500	ML5/ HPA005785	Biotinylated goat anti-mouse AP and Envision goat anti-rabbit HRP	
**ALDH1**	CC1 ^#^	121 °C/6 min	1:200	44/ALDH	Envision goat anti-mouse	
**Ep-CAM**	Citrate	121 °C/8 min	1:500	VU-1D9	Envision goat anti-mouse	
**BMI1**	CC1	121 °C/6 min	1:500	F6	Biotinylated goat anti-mouse (1/1000) (JL) ⊥	
**CD10 ^#^**	CC1	(clone SP67), Ventana Benchmark Ultra system				
**PD-L1 ^#^**	(clone SP263), Ventana Benchmark Ultra system, previously described [54]					
**FOXP3 and PD-1**	(Clones 236A/E7 and J116, respectively), previously described [56]					
**Vimentin ^#^/E-Cadherin**	(Clones V9 and EP700Y, respectively), previously described [52]					
**SKP2, p21, and p27**	(Clones D3G5, 12D1, and F-8 respectively), previously described [55]					

* Indicated temperature is set point (SP)1, while SP2 is always 95 °C for 10 min. ^#^ Obtained from Ventana, AZ, USA. ⊥ Obtained from Jackson Laboratories, USA.

## Data Availability

The data generated in this current study are included in this published article (and its Appendix A), otherwise available from the corresponding author on reasonable request.

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
