# Peer review of "Profiling of Breast Cancer Stem Cell Types/States Shows the Role of CD44hi/CD24lo-ALDH1hi as an Independent Prognostic Factor After Neoadjuvant Chemotherapy"

_ijms, 2025, doi:10.3390/ijms26178219_

Round 1

Reviewer 1 Report

Comments and Suggestions for Authors

This is a very important and interesting article. Authors provided analyse of multiple markers exist for breast cancer stem cells (CSCs).

There were the phenotypes of various CSC types and/or states in the article.

Authors investigated the correlation of different

CSC type/state with markers of immune evasion, cell proliferation, and EMT.

However, the article does not describe the patients included in the study.

The chapter on materials and methods does not describe all markers in detail (no.)

A good Immunohistochemistrical analysis was performed, but it would be more interesting to look at these markers using cytometric analysis.

It is known that populations of cells - Hi and low are more often distinguished in cytometric studies.

Comments on the Quality of English Language

The English language Is appropriate and understandable.

Author Response

1. This is a very important and interesting article. The authors provided an analysis of multiple markers that exist for breast cancer stem cells (CSCs). There were the phenotypes of various CSC types and/or states in the article. The authors investigated the correlation of different CSC types/states with markers of immune evasion, cell proliferation, and EMT.

Response to comment 1

  • Thank you.

2. The article does not describe the patients included in the study.

Response to comment 2

  • Thank you. We have now included a patient's characteristics section in the results to address this comment (lines 78-82, highlighted in yellow).

3. The chapter on materials and methods does not describe all markers in detail (no.)

Response to comment 3

  • Thank you, we have now expanded and improved this section to address this comment (lines 456-461 and table 1, highlighted in yellow).

4. A good immunohistochemical analysis was performed, but it would be more interesting to look at these markers using cytometric analysis. It is known that populations of cells - Hi and low are more often distinguished in cytometric studies.

Response to comment 4

  • Thank you. Indeed, hi and lo populations will be better distinguished with flow cytometry. However, flow cytometric studies require a single suspension of digested tumor tissues, which is not available for these archived samples. In addition, while certain CSCs markers like CD44/CD24, ALDH, Ep-CAM and CD10 can be studied in flow cytometry, other nuclear markers like BMI1, SKP2, p21, and p27 might be more challenging to study using flow cytometry. Another challenge for flow cytometry is the fact that cancer cells cannot be recognized appropriately as compared to tissue sections in immunohistochemistry, which can be easily distinguished using light microscopy (using morphology like pleomorphic nucleus, visible nucleolus, etc., which is within the expertise of anatomical pathologists). Even if epithelial markers like Ep-CAM, were used, some cancer cells will be missed (we have shown that Ep-CAM is negative in certain tumors), which could be mixed with normal-looking mammary epithelial cells that sometimes co-exist in tumor tissues.

Reviewer 2 Report

Comments and Suggestions for Authors

The authors have shown a very detailed / solid study. It is very well done. The conclusion section is suggested to be shown. A statement of the shortage or drawback of current study is suggested to be shown in the end of the discussion. One question about this study may be asked. It is  about the drugs or treatments used in Neoadjuvant chemotherapy. If different drugs or treatments are  used in neoadjuvant chemotherary, the expression of  ALDHhi/CD44hi/CD24lo are still the same to that  shown in this study? Is there any dependence? I guess the authors can have that discussion in the manuscript.

Author Response

1. The authors have shown a very detailed/solid study. It is very well done.

Response to comment 1

  • Thank you

2. The conclusion section is suggested to be shown.

Response to comment 2

  • Thank you. We have added a conclusion section as suggested by the reviewer (lines 482-495, highlighted in yellow).

3. A statement of the shortage or drawback of the current study is suggested to be shown at the end of the discussion.

Response to comment 3

  • Thank you. The limitations of the study is now highlighted at the end of the discussion (lines 426-433, highlighted in yellow).

4. One question about this study may be asked. It is about the drugs or treatments used in Neoadjuvant chemotherapy. If different drugs or treatments are used in neoadjuvant chemotherapy, will the expression of CD44hi/CD24lo-ALDHhi be the same as that shown in this study? Is there any dependence? I guess the authors can have that discussion in the manuscript.

Response to comment 4

  • We have now added the neoadjuvant chemotherapy used in this study. We have shown that most patients received Adriamycin (doxorubicin) with Cyclophosphamide (AC), which was taken alone, with fluorouracil (part of the FAC regimen) or with Docetaxel (AC+docetaxel). We have further performed subgroup analysis, and we have demonstrated that breast cancers positive for CD44hi/CD24lo-ALDHhi have significantly worse prognosis in patients who received AC or FAC neoadjuvant chemotherapy (new supplementary figure 1 is now added).